# The Effects of the Structural and Acoustic Parameters of the Skull Model on Transcranial Focused Ultrasound

**DOI:** 10.3390/s21175962

**Published:** 2021-09-05

**Authors:** Hao Zhang, Yanqiu Zhang, Minpeng Xu, Xizi Song, Shanguang Chen, Xiqi Jian, Dong Ming

**Affiliations:** 1Laboratory of Neural Engineering and Rehabilitation, Department of Biomedical Engineering, College of Precision Instruments and Optoelectronics Engineering, Tianjin University, Tianjin 300072, China; haozhang_bme@tju.edu.cn (H.Z.); xmp52637@tju.edu.cn (M.X.); shanguang_chen@126.com (S.C.); 2School of Biomedical Engineering and Technology, Tianjin Medical University, Tianjin 300070, China; tmu_zyq@126.com (Y.Z.); jianxiqi@tmu.edu.cn (X.J.); 3Academy of Medical Engineering and Translational Medicine, Tianjin University, Tianjin 300072, China; songxizi@tju.edu.cn; 4National Key Laboratory of Human Factors Engineering, China Astronaut Research and Training Center, Beijing 100094, China

**Keywords:** transcranial focused ultrasound, structural parameter, acoustic parameter, numerical simulation

## Abstract

Transcranial focused ultrasound (tFUS) has great potential in brain imaging and therapy. However, the structural and acoustic differences of the skull will cause a large number of technical problems in the application of tFUS, such as low focus energy, focal shift, and defocusing. To have a comprehensive understanding of the skull effect on tFUS, this study investigated the effects of the structural parameters (thickness, radius of curvature, and distance from the transducer) and acoustic parameters (density, acoustic speed, and absorption coefficient) of the skull model on tFUS based on acrylic plates and two simulation methods (self-programming and COMSOL). For structural parameters, our research shows that as the three factors increase the unit distance, the attenuation caused from large to small is the thickness (0.357 dB/mm), the distance to transducer (0.048 dB/mm), and the radius of curvature (0.027 dB/mm). For acoustic parameters, the attenuation caused by density (0.024 dB/30 kg/m^3^) and acoustic speed (0.021 dB/30 m/s) are basically the same. Additionally, as the absorption coefficient increases, the focus acoustic pressure decays exponentially. The thickness of the structural parameters and the absorption coefficient of the acoustic parameters are the most important factors leading to the attenuation of tFUS. The experimental and simulation trends are highly consistent. This work contributes to the comprehensive and quantitative understanding of how the skull influences tFUS, which further enhances the application of tFUS in neuromodulation research and treatment.

## 1. Introduction

Transcranial focused ultrasound (tFUS) is a non-invasive, localized, non-ionizing technique with applications in particular in oncology and neurosurgery. It has been investigated for over 60 years, with seminal studies by Lynn and Fry establishing the potential of applying ultrasound to cerebral tissue [1,2,3]. Depending on stimulation parameters, tFUS is able to determine a wide spectrum of effects, ranging from suppression or facilitation of neural activity to tissue ablation [4]. According to the acoustic intensity of the focal area, tFUS can be divided into low-intensity focused ultrasound (LIFU) and high-intensity focused ultrasound (HIFU).

Numerous experimental studies have demonstrated that LIFU is an emerging tool for non-invasive neuromodulation. It can transmit low-intensity ultrasound through the skull to temporarily and safely modulate regional brain activity [5,6,7,8]. In 2010, Tufail used targeted LIFU to stimulate neuronal activity in the intact hippocampus. The experimental results show that ultrasound-stimulated neuronal activity was sufficient to evoke mice’s motor behaviors [9]. In 2013, Deffieux demonstrated the feasibility of using LIFU stimulation to causally modulate behavior in the awake nonhuman primate brain for the first time [10]. In 2014, Legon probed the influence of targeted LIFU to the human primary somatosensory cortex on sensory-evoked brain activity and sensory discrimination abilities. Electroencephalographic recordings showed that LIFU can be used to focally modulate human cortical function [11]. LIFU can not only achieve neuromodulation in normal animals and humans, but also has shown positive and effective effects in the prevention and treatment of neurological diseases such as epilepsy [12], Parkinson’s disease [13], and depression [14].

As a non-invasive tumor treatment technology, HIFU has been widely used in the treatment of solid tumors and has broad application prospects [15]. In initial trials in three glioblastoma patients in 2010, multiple HIFU exposures were applied up to the maximum acoustic power available. However, the skull makes ultrasound delivery difficult or even impossible and thus seemed to not achieve thermal coagulation [16]. In 2014, Coluccia applied HIFU under MR imaging guidance, and the partial tumor ablation could be achieved. This work is the first to demonstrate the feasibility of using tFUS to safely ablate substantial volumes of brain tumor tissue. Unfortunately, the applications of tFUS in the brain have been inhibited by the skull, as well as the inability to target and monitor treatments. Over the years, with the goal of understanding the impact of the skull on tFUS, scholars have done a lot of work, including mapping the geometry of the skull [17], exploring the acoustic parameters of the skull [18,19], trying skull phantom [20], and exploring the effect of skull thickness and density on tFUS [21,22,23]. Although previous studies have achieved certain results, these works have not been systematically and comprehensively studied on the influence of the skull on tFUS.

Therefore, the present study aims to investigate this aspect. This article mainly consists of two parts: the first part focuses on the quantitative description of the influence of structural parameters (thickness, curvature radius, and distance to transducer) on tFUS, while the second part contains an accuracy assessment of the impact of acoustic parameters (density, acoustic speed, and absorption coefficient).

## 2. Materials and Methods

### 2.1. Mathematical Equation for Acoustic Field

Based on the self-programming simulation, we have already accurately reproduced the tFUS propagation through the skull [24,25]. Throughout the paper, the nonlinear Westervelt equation is used to describe acoustic fields in the simulation [26,27]. It incorporates the effects of nonlinearity, attenuation, reflection, refraction, and scattering of waves, and can be written as
(1)$$\nabla^{2}p-\frac{1}{c^{2}}\frac{\partial^{2}p}{\partial t^{2}}+\frac{\delta }{c^{4}}\frac{\partial^{3}p}{\partial t^{3}}+\frac{\beta }{\rho c^{4}}\frac{\partial^{2}p^{2}}{\partial t^{2}}=0$$

In this equation, p is acoustic pressure, c is acoustic speed, t is time, ρ is medium density, δ is the thermoviscous diffusivity, β is the nonlinearity coefficient. The equation is calculated by the finite-difference time-domain method (FDTD) in 3D. Since the center frequency of the ultrasound is 0.9 MHz, the simulation space step is set to one-eighth of the ultrasonic wavelength in water (0.2 mm), and the time step is 10 ns.

In the COMSOL Multiphysics 5.4 (COMSOL Inc., Stockholm, Sweden) simulation part, this paper uses the Helmholtz equation to calculate the acoustic field
(2)$$\frac{\partial }{\partial r}\left[-\frac{r}{\rho_{c}}\left(\frac{\partial p}{\partial r}\right)\right]+r\frac{\partial }{\partial r}\left[-\frac{1}{\rho_{c}}\left(\frac{\partial p}{\partial z}\right)\right]-\left[{\left(\frac{\omega }{c}\right)}^{2}\right]\frac{rp}{\rho_{c}}=0$$

In this equation, r is radial coordinate, z. is axial coordinate, p is acoustic pressure, ω is angular frequency, $\rho_{c}$ is density, and c is acoustic speed. The remaining settings of the two simulations are exactly the same. Numerical simulations are performed on the OMNISKY MultiGPU CSC4450 server platform with an internal configuration of dual Xeon E5-2697 V3 processor (2.60 GHz).

### 2.2. Numerical Simulation Model

Figure 1 schematically shows the numerical simulation model, including the transducer, skull, and water, which size of the numerical domain is 100 × 100 × 100 mm. The Mur first-order absorbing boundary condition was used at the boundary to avoid the reflection of ultrasound [28]. Among them, we first focus on the thickness of the skull (T:2:4:22 mm), the radius of curvature (R:60:10:100 mm), and the distance to transducer (D:11:1:20 mm) according to different positions (such as temporal bone, occipital bone, parietal bone, etc.) of the skull [17]. According to previous studies, skull density, acoustic speed, and absorption coefficient trended toward a reduction with increasing age [29,30]. According to the suggestion in the literature, the density of the skull ($\rho :1700:30:2000\mathrm{kg}/m^{3}$) [19], the acoustic speed (c:2000:300:5000 m/s) [18], and the absorption coefficient (α:200:480:5000 dB/m) [31] are also considered in the present study [32,33]. The physical properties of the objects are shown in Table 1.

### 2.3. Experimental Setup

Since the acoustic properties of the acrylic plate (such as density, sound speed, and attenuation coefficient, etc.) are close to those of the skull, the acrylic plate phantom mimics the skull [18]. Based on computer numerical control (CNC) technology, acrylic plates with different thicknesses and curvature radius are designed (Figure 2). The experimental schematic and device diagram are shown in Figure 3. The ultrasound transducer is a custom designed single-element piezoelectric ceramic focused transducer (Fluids Engineering Laboratory, Department of Mechanical Engineering, the University of Tokyo, Japan), which has a center frequency of 0.9 MHz, an opening diameter of 100 mm, a focal length of 80 mm, and a central hole diameter of 50 mm. The power amplifier (AG1021 T&C, USA) is used to amplify the signal induced by the signal generator (33522A Agilent, Santa Clara, CA, USA), and further drive the transducer using the continuous wave (CW) mode to generate a 10s continuous sine wave. The ultrasound field around the geometrical focus is measured by polyvinylidene fluoride (PVDF) membrane needle hydrophone (NCS-1, Beijing, China) mounted on a three-axis stage (Sigma Koki, Tokyo, Japan). All experiments are carried out in a degassed water tank.

## 3. Results

The results section is divided into three parts. First, the consistency between the simulation and experiment is explored. The second and third parts are the influence of skull model structural parameters and acoustic parameters on tFUS.

### 3.1. Consistency between Simulation and Experiment

#### 3.1.1. Transducer Surface Pressure

To ensure consistency of the initial conditions of the experiment and simulation, we first use a hydrophone close to the transducer in pure water to measure its surface sound pressure. Then, the acoustic pressures at 12 points distributed on the surface of the transducer according to the clock dial are measured under different excitation conditions (Figure 4a), and its average value is used as the initial acoustic pressure for the simulations. Figure 4b,c show the acoustic pressure value at each point and overall trends under different excitation power. It can be seen that the transducer surface pressure increased with increasing excitation power, and fluctuated slightly.

#### 3.1.2. Acoustic Pressure Field Comparison

Under the same initial conditions (excitation power 1 w), we have explored the influence of the presence of skull model (T8 mm, R60 mm, D10 mm) on tFUS. As shown in Figure 5a–d, both simulation and experiment accurately focus at the geometric focal point 80 mm under without skull model condition. The influence of the acrylic plate as an elastic body on the distortion of focus is shown in Figure 5e–h; while Figure 5e is a self-programming acoustic pressure field, the Figure 5f is COMSOL acoustic pressure field. In the presence of the bone plate, the experimental and simulated focal points have shifted about 2–3 mm from the geometric focal point to the transducer side, which is consistent with the results of previous studies [35]. The attenuation caused by the acrylic plate was very strong, with 56.22%, 57.65%, and 56.87% for self-programming, experimental, and COMSOL, respectively (Table 2), which is very similar to the results in the literature [19,36]. In other words, the existence of the skull model will significantly change the focus position and the focusing region energy.

### 3.2. The Influence of the Skull Model Structural Parameter

Under the condition of ensuring the rest of the parameters remained unchanged, we have changed the thickness of the skull model, the radius of curvature, and the distance to transducer one by one.

#### 3.2.1. Thickness

The influence of acrylic plate thickness on ultrasound propagation is examined next. The fixed distance of the acrylic plate (Excitation power: 1 w, R: 60 mm) from the open end of the transducer (D) is 10 mm and its thickness grew from 2 mm to 22 mm. Figure 6a displays the linear fitting results where different colors represent different numerical simulations and experiments. The self-programming curve is shown in blue; the experimental and COMSOL curves are in orange and green, respectively. The numerical simulation and experimental results show that the focus acoustic pressure decreased with increasing acrylic plate thickness. The focus acoustic pressure in self-programming fell by 81.13% compared with 78.84% in experimental, and by 83.04% in COMSOL. According to the fitting formula, we calculated the focus acoustic attenuation value for a 1mm increase in thickness. Compared with the initial acoustic pressure, the focus sound pressure will be reduced by 0.357 dB, 0.378 dB, and 0.353 dB, respectively.

#### 3.2.2. Curvature Radius

We then studied the influence of an acrylic plate curvature radius on the propagation of ultrasonic waves. The distance of the acrylic plate (excitation power: 1 w, T: 8 mm) from the open end of the transducer (D) is 10 mm and its curvature radius grew from 60 mm to 100 mm. Figure 6b shows the linear fitting curves of the three results. The numerical simulation and experimental results show that the focus acoustic pressure decreased with increasing acrylic plate curvature radius. The focus acoustic pressure in self-programming fell by 12.41% compared with 14.83% in experimental, and by 5.24% in COMSOL. From the fitting results, the focus acoustic pressure will decrease by 0.027 dB, 0.032 dB, and 0.011 dB for every 1 mm increase in curvature radius, respectively.

#### 3.2.3. Distance to Transducer

In the next step, the influence of an acrylic plate distance to transducer has been regarded as a research target in this part. We use a three-axis stage to change the acrylic plate (excitation power: 1 w, T: 8 mm, R: 60 mm) distance from the open end of the transducer (D: 11 mm–20 mm). Figure 6c shows the linear fitting curves of the three results. The numerical simulation and experimental results show that the focus acoustic pressure decreased with increasing distance to the transducer. The focus acoustic pressure in self-programming fell by 5.53% compared with 10.23% in experimental, and by 2.15% in COMSOL. According to the fitting formula, the focus acoustic pressure will decrease by 0.048 dB, 0.089 dB, and 0.019 dB for every increase of 1 mm between the acrylic plate and transducer.

The linear fitting results under different conditions are shown in Table 3, which mainly include the coefficient of determination (R-square), the fitting formula, the correlation coefficient between fitting curves, and the attenuation of unit distance. X is the independent variable, representing the variation of thickness, radius of curvature, and distance from the transducer, respectively. Y is the focus acoustic pressure. In addition, the higher the Pearson correlation coefficient, the higher the degree of agreement between the curve and the self-programming fitting curve. It can be seen from the slope of the fitted curve and the attenuation results per unit distance that the importance of the impact is: (1) thickness; (2) distance to the transducer; (3) radius of curvature. The self-programming results show that the attenuation of the thickness of the skull model (0.357 dB/mm) is about 7.44 times the distance to the transducer (0.048 dB/mm) and about 13.22 times the radius of curvature (0.027 dB/mm).

### 3.3. The Influence of the Skull Model Acoustic Parameter

We then explore the influence of the skull model acoustic parameters on tFUS based on self-programming and COMSOL. Additionally, corresponding structural parameters are set as excitation power 1 w, T8 mm, R60 mm, D10 mm. First, we change all three parameters of the linear interval simultaneously. Then, to evaluate the effect of varied acoustic parameters, we vary only one parameter at a time, while keeping others constant.

#### 3.3.1. Comprehensive Influence of Three Parameters

In the first part, we take the values of density ρ (1700:30:2000 kg/m^3^), acoustic speed c (2000:300:5000 m/s), and absorption coefficient α (200:480:5000 dB/m) in linear intervals, and change these three acoustic parameters simultaneously. Figure 7a,c are self-programming and COMSOL acoustic pressure fields and the variation trend of focus acoustic pressure under different parameters are displayed in Figure 7b,d. Figure 7e shows the histogram of the change in the focus acoustic pressure for the two simulation methods under different acoustic parameters. Our results indicate that the declining trend of the focus acoustic pressure gradually becomes smooth as three parameters are increased at the same time. It needs to be noted here that it is difficult to form a distinguishable focal region after ρ 1880 kg/m^3^, c 3800 m/s, and α 3080 dB/m. At this point, the self-programming acoustic pressure decrease by 94.41%, and COMSOL decline by 94.68%. To determine which acoustic parameter is most influential for the determination of the attenuation of the acoustic pressure, we have varied parameters one by one.

#### 3.3.2. Influence of a Single Coefficient: Density

In the second part, skull density ρ is increased from 1700 to 2000 kg/m^3^ at 30 kg/m^3^ intervals, while maintaining the fixed acoustic speed c 3500 m/s and absorption coefficient α 2600 dB/m. Figure 8a,c are self-programming and COMSOL acoustic pressure fields and the variation trend of focus acoustic pressure under different densities are displayed in Figure 8b,d. Figure 8e,f show the fitted curves and histograms of the change in focus acoustic pressure for the two simulation methods at different density conditions. The results show that with the increase of density, the focus acoustic pressure shows a slight downward trend. The self-programming acoustic pressure decrease by 3.12% in total, and 13.02% in the COMSOL. From the linear fitting diagram, we calculated the focus acoustic attenuation value for every 30 kg/m^3^ increase in density. Compared with the initial acoustic pressure, the focus sound pressure will be reduced by 0.024 dB, and 0.115 dB, respectively.

#### 3.3.3. Influence of a Single Coefficient: Acoustic Speed

The third part is the effect of acoustic speed, skull acoustic speed c is increased from 2000 to 5000 m/s at 300 m/s intervals, while maintaining the fixed density ρ 1850 kg/m^3^ and absorption coefficient α 2600 dB/m. Figure 9a,c are self-programming and COMSOL acoustic pressure fields and the variation trend of focus acoustic pressure under different acoustic speed are displayed in Figure 9b,d. Figure 9e,f show the fitted curves and histograms of the change in focus acoustic pressure for the two simulation methods at different density conditions. The results showed that with the increase of acoustic speed, the focus acoustic pressure shows a significant downward trend. The self-programming sound pressure decrease by 25.01% in total, and 50.14% in the COMSOL. From the linear fitting diagram, we calculated the focus acoustic attenuation value for every 30 m/s increase in acoustic speed. Compared with the initial acoustic pressure, the focus sound pressure will be reduced by 0.021 dB, and 0.055 dB, respectively.

#### 3.3.4. Influence of a Single Coefficient: Absorption Coefficient

The final portion of the acoustic parameter is the impact of the absorption coefficient. Skull absorption coefficient α is increased from 200 to 5000 dB/m at 480 dB/m intervals, while maintaining the fixed density ρ 1850 kg/m^3^ and acoustic speed c 3500 m/s. Figure 10a,c are self-programming and COMSOL acoustic pressure fields and the variation trend of focus acoustic pressure under different absorption coefficients are displayed in Figure 10b,d. Figure 10e,f show the fitted curves and histograms of the change in focus acoustic pressure for the two simulation methods at different density conditions. The most spectacular downward trend is observed for acoustic pressure as the absorption coefficient increases and it is difficult to form a distinguishable focal region after α 3080 dB/m. At this point, the self-programming acoustic pressure decrease by 93.66%, and COMSOL decline by 89.64%. The manuscript calculates the slope of each segment at an interval of 480 dB/mm based on the fitting formula, as shown in Figure 10e. The results show that after α 3080 dB/m, the slope change becomes significantly smaller and the attenuation of sound pressure tends to be flat.

Table 4 shows the fitting results under the three conditions. Where x is the independent variable, representing the variation of density, acoustic speed, and absorption coefficient, respectively. Y is the focal sound pressure. The results show that the effect of the acoustic absorption coefficient on acoustic parameters is the most critical. Among them, the self-programming results show that the attenuation caused by the acoustic speed of the skull model (0.021 dB/mm) is basically the same as the attenuation caused by the density (0.024 dB/mm).

## 4. Discussion

In this study, we have analyzed the effects of skull model structural and acoustic parameters on tFUS based on simulation and experiment. These skull parameters will vary with the patient’s age, physiological state, and ultrasound incident acoustic window. For variation tendency and the importance ranking, the self-programming results are consistent with the experimental and COMSOL results. To an extent, with the increase of skull model thickness, radius of curvature, distance to transducer, density, acoustic speed, and absorption coefficient, the focus acoustic pressure of the self-programming, experiment, and COMSOL all showed a decreasing tendency of varying degree. Specifically, as the thickness, radius of curvature, and distance to transducer increase by 1mm, the self-programming results show that the focus acoustic pressure drops by 0.357 dB, 0.027 dB, and 0.048 dB, respectively. As the density and acoustic speed increase by 30 kg/m^3^ and 30 m/s, the self-programming results show that the focus acoustic pressure drops by 0.024 dB and 0.021 dB, respectively. In addition, as the absorption coefficient increases at an interval of 480 dB/m, the focus acoustic pressure drops from sharp to gentle. Our results indicate that the thickness and absorption coefficient play the most important role in the skull model structural and acoustic parameters.

Nevertheless, we are still seeing some reasonable differences between self-programming simulation, experiment, and COMSOL simulation. After comprehensive consideration, potential major reasons for the observed acoustic pressure field differences between self-programming and COMSOL include the following three considerations. First of all, the acoustic field calculation formulas of the two simulations are different, one is Westervelt and the other is Helmholtz. Westervelt equation is used to describe nonlinear propagation in many cases, while Helmholtz equation is mainly used to describe linear propagation. Secondly, the acoustic wave propagation equation of self-programming additionally considers nonlinear factors. The acoustic phenomena are fundamentally nonlinear, and can be described by an approximate linearity when the ultrasound amplitude energy is low, and the nonlinear effect becomes progressively larger as the ultrasound amplitude increases. At last, for the numerical calculation method of the acoustic wave propagation equation, self-programming uses the FDTD method, while COMSOL uses the finite element method (FEM). Both FDTD and FEM are electromagnetic wave simulation algorithms. FDTD is faster in calculation and has average accuracy, FEM is average in calculation and has higher accuracy. About the unavoidable differences between experiments and numerical simulations, these are likely caused by simulation parameters that could be mismatched with respect to the true parameters. Besides, the energy inhomogeneity of the transducer surface is ignored here in the numerical simulation. And due to the spherical shape of the transducer surface. There is a non-right angle between the hydrophone and the transducer, and the pointing of the hydrophone may have an effect on the measurement results.

In addition, this study uses homogeneous acrylic plates to simulate the human skull (mainly cortical bone), ignoring the trabecular bone structure. A smooth and ideal homogeneous acrylic plate may not have the problems of scattering, aberration, and sound field distortion caused by the heterogeneous structure of the real skull, which is different from the real situation. In previous studies, we have used self-programming simulation to build a head model based on real human skull CT data, and further simulated the propagation of tFUS in the skull and brain tissue. The main purpose of this article is to explore the influence of skull model parameters on tFUS one by one, and to screen out the most influential parameters, so it is relatively reasonable to use acrylic board here. With the development of phase-controlled transducer technology, researchers have been able to achieve precise intracranial focusing. However, it generally requires a time-reversal method to obtain the phase information of each element, and the phased array system is generally expensive to manufacture. When exploring the influence of a certain parameter, the advantages of phase-controlled transducers are not so obvious. In this paper, based on a single-element transducer, quantitatively and systematically explored the impact of changes in skull parameters on tFUS due to the patient’s age, physiological state, and ultrasound incident acoustic window.

In future research, we will establish the patient’s heterogeneous head model based on its head CT (Computed Tomography)/MRI (Magnetic Resonance Imaging) data, and further explore the impact of heterogeneous skull parameter changes on tFUS, especially the problems of scattering and aberration. Related research will help the application of tFUS in neuromodulation research and treatment.

## 5. Conclusions

The current study comprehensively explores the effects of the structural and acoustic parameters of the skull model on tFUS. It contains one experimental study and two simulation studies. The results show that the focus acoustic pressure showed different degrees of decreasing trend with the increase of structural parameters or acoustic parameters. Both simulation methods can accurately fit the acoustic pressure variation trend, and the experimental and simulation fitting curves are highly correlated. This study provides a reference for the application of tFUS in neuromodulation and therapy.

## Figures and Tables

**Figure 1 sensors-21-05962-f001:**
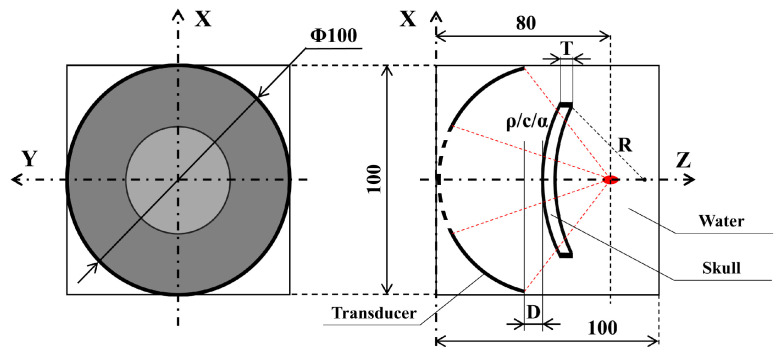
Schematic diagram of the numerical simulation model.

**Figure 2 sensors-21-05962-f002:**
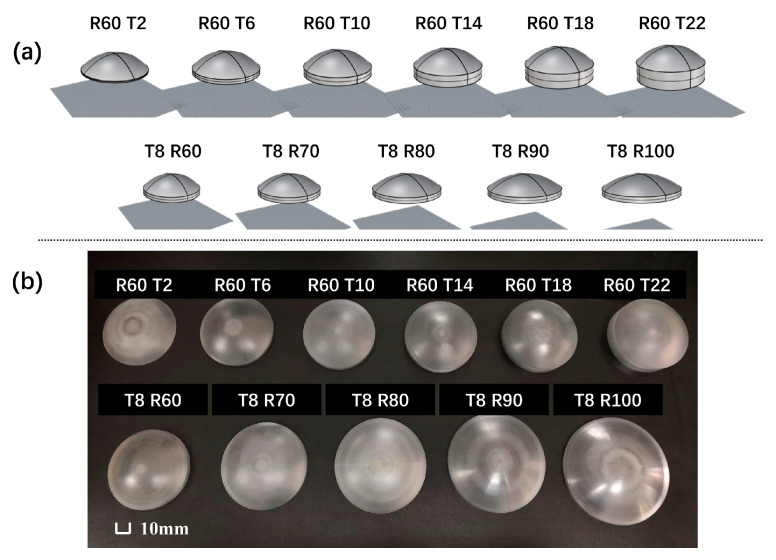
(**a**) CNC-printed prototypes; (**b**) physical map of the acrylic plate (unit: mm).

**Figure 3 sensors-21-05962-f003:**
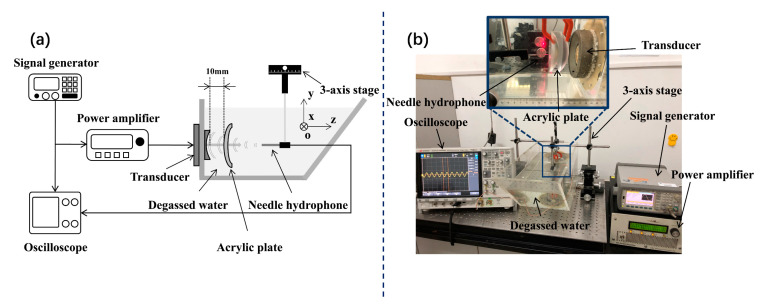
(**a**) Schematic diagram of the experimental setup; (**b**) physical map of the experimental setup.

**Figure 4 sensors-21-05962-f004:**
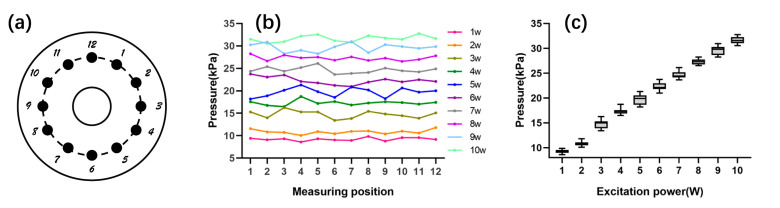
Transducer surface acoustic pressure; (**a**) schematic diagram of sampling points; (**b**) acoustic pressure at each point under different excitation conditions; (**c**) the variation trend of surface acoustic pressure.

**Figure 5 sensors-21-05962-f005:**
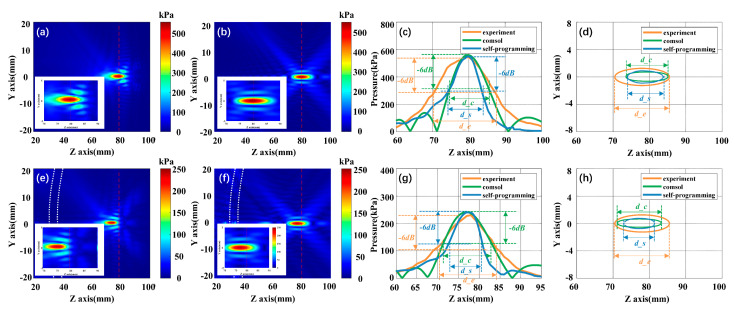
Acoustic pressure field comparison. (**a**) Focus acoustic pressure field of self-programming under pure water; (**b**) focus acoustic pressure field of COMSOL under pure water; (**c**) acoustic axis pressure curves comparison under pure water; (**d**) focus regions comparison chart under pure water; (**e**) self-programming focus acoustic pressure field in the presence of an acrylic plate; (**f**) COMSOL focus acoustic pressure field in the presence of an acrylic plate; (**g**) acoustic axis pressure curves comparison chart in the presence of acrylic plate; (**h**) focus regions comparison in the presence of acrylic plate.

**Figure 6 sensors-21-05962-f006:**
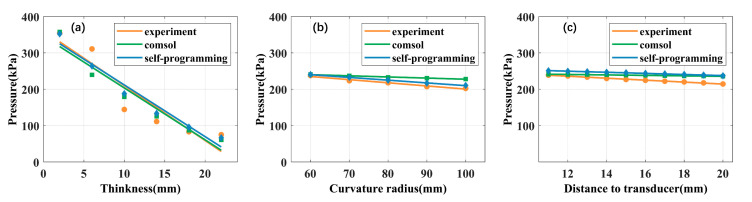
Trend chart of acoustic pressure changes under different conditions. (**a**) The trend of acoustic pressure changes due to changes in skull model thickness; (**b**) the changing trend of acoustic pressure due to changes in the radius of curvature of the skull model; (**c**) the changing trend of acoustic pressure due to the change of the distance between the skull model and the transducer.

**Figure 7 sensors-21-05962-f007:**
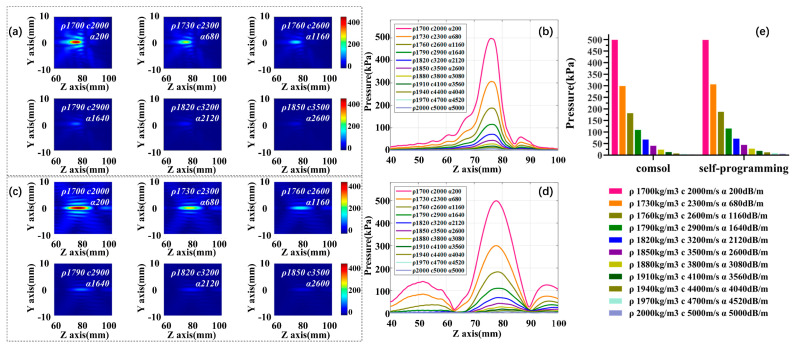
The effect of three factors varied simultaneously on focus acoustic pressure. (**a**) Self-programming focus acoustic pressure field; (**b**) self-programming acoustic axis acoustic pressure curve; (**c**) COMSOL focus acoustic pressure field; (**d**) COMSOL acoustic axis acoustic pressure curve; (**e**) histogram of focus acoustic pressure variation for both simulations.

**Figure 8 sensors-21-05962-f008:**
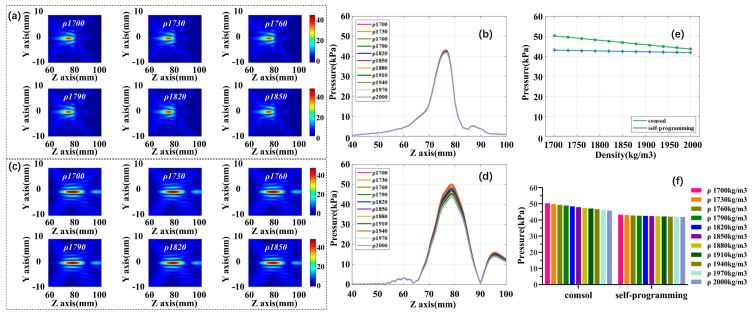
The effect of density varied on focus acoustic pressure. (**a**) Self-programming focus acoustic pressure field; (**b**) self-programming acoustic axis acoustic pressure curve; (**c**) COMSOL focus acoustic pressure field; (**d**) COMSOL acoustic axis acoustic pressure curve; (**e**) focus acoustic pressure variation fitting curves for both simulations; (**f**) histogram of focus acoustic pressure variation for both simulations.

**Figure 9 sensors-21-05962-f009:**
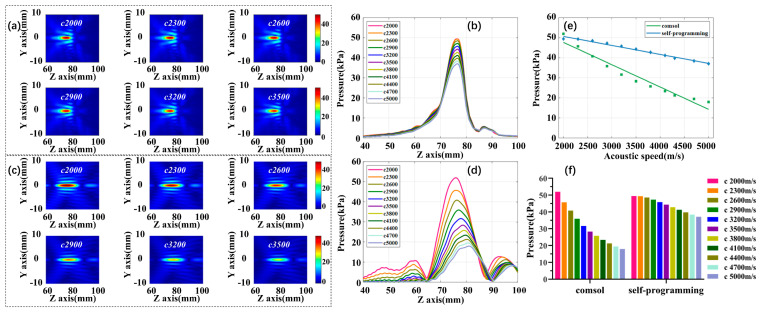
The effect of acoustic speed varied on focus acoustic pressure. (**a**) Self-programming focus acoustic pressure field; (**b**) self-programming acoustic axis acoustic pressure curve; (**c**) COMSOL focus acoustic pressure field; (**d**) COMSOL acoustic axis acoustic pressure curve; (**e**) focus acoustic pressure variation fitting curves for both simulations; (**f**) histogram of focus acoustic pressure variation for both simulations.

**Figure 10 sensors-21-05962-f010:**
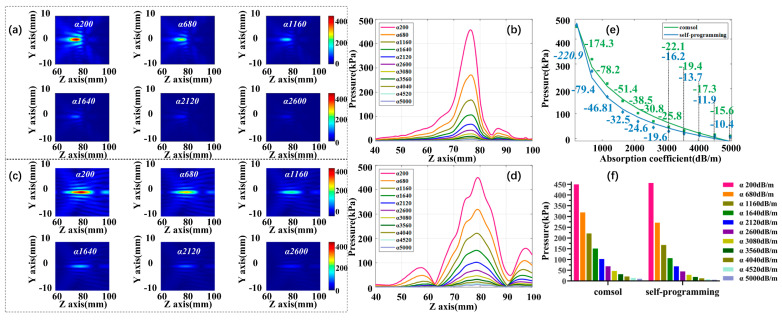
The effect of absorption coefficient varied on focus acoustic pressure. (**a**) Self-programming focus acoustic pressure field; (**b**) self-programming acoustic axis acoustic pressure curve; (**c**) COMSOL focus acoustic pressure field; (**d**) COMSOL acoustic axis acoustic pressure curve; (**e**) focus acoustic pressure variation fitting curves for both simulations; (**f**) histogram of focus acoustic pressure variation for both simulations.

**Table 1 sensors-21-05962-t001:** Simulation parameters table of water and acrylic plate.

	Density (kg/m^3^)	Sound Speed (m/s)	Attenuation Coefficient (Np/m/MHz)	Nonlinearity Coefficient (Only Self-Programming)
Water	998.2 [18]	1482 [18]	0.02 [19]	3.5 [24]
Acrylic	1203 [18]	2600 [18]	18 [34]	4.4 [24]

**Table 2 sensors-21-05962-t002:** Comparison of pure water and transcranial results.

		Self-Programming	Experiment	COMSOL
Pure water	Focus position	(79.9,0)	(80,0)	(79.8,0)
	Focus pressure (kPa)	549.3	543.8	559.7
	−6 dB length (mm)	9.8(d_s)	16(d_e)	12.2(d_c)
Transcranial	Focus position	(77.2,0)	(78,0)	(76.9,0)
	Focus pressure (kPa)	240.5	230.3	241.4
	−6 dB length (mm)	8.3(d_s)	15(d_e)	11.5(d_c)

**Table 3 sensors-21-05962-t003:** Structural parameters fitting results.

	Fit Parameters	Self-Programming	Experiment	COMSOL
Thickness	R-square	0.948	0.862	0.936
	Fitting formula	y = −14.2x + 353.6	y = −15.09x + 360.7	y = −14.26x + 346
	Pearson correlation coefficient	/	0.956	0.996
	Unit attenuation (dB/mm)	0.357	0.378	0.353
Curvature radius	R-square	0.998	0.972	0.984
	Fitting formula	y = −0.755x + 285.3	y = −0.868x + 286.9	y = −0.315x + 258.9
	Pearson correlation coefficient	/	0.989	0.994
	Unit attenuation (dB/mm)	0.027	0.032	0.011
Distance to transducer	R-square	0.997	0.995	0.977
	Fitting formula	y = −1.521x + 267.8	y = −2.695x + 268	y = −0.639x + 248.2
	Pearson correlation coefficient	/	0.983	0.986
	Unit attenuation (dB/mm)	0.048	0.089	0.019

**Table 4 sensors-21-05962-t004:** Acoustic parameters fitting results.

	Fit Parameters	Self-Programming	COMSOL
Density	R-square	0.980	0.999
	Fitting formula	y = −0.004x + 50.19	y = −0.022x + 87.39
	Pearson correlation coefficient	/	0.993
	Unit attenuation (dB/30 kg/m^3^)	0.024	0.115
Acoustic speed	R-square	0.985	0.953
	Fitting formula	y = −0.004x + 59.65	y = −0.011x + 69.99
	Pearson correlation coefficient	/	0.949
	Unit attenuation (dB/30 m/s)	0.021	0.055
Absorption coefficient	R-square	0.990	0.986
	Fitting formula	y = 2993x^−0.218^ − 418.2	y = −3701x^0.032^ + 4844
	Pearson correlation coefficient	/	0.991

## Data Availability

Not applicable.

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
