# Peer review of "The Effects of the Structural and Acoustic Parameters of the Skull Model on Transcranial Focused Ultrasound"

_sensors, 2021, doi:10.3390/s21175962_

Round 1

Reviewer 1 Report

The authors addressed important issue in the influence of structural (e.g., thickness, radius of curvature and distance from the transducer) and acoustic (density, acoustic speed and absorption coefficient) parameters of a skull model with similar morphology and acoustic characteristics to a real one. The obtained results were critiaclly evaluated, presented and well discussed. The findings will be of importance in further clinical practice.

Author Response

Dear reviewer:

On behalf of my co-authors, thank you very much for your approval of the article. In addition, we have thoroughly checked the article for grammar and spelling errors, and hope that the updated article will meet your expectations.

We will be looking deeper into this direction and hope to communicate with you more in the future!

Good luck!

Reviewer 2 Report

The authors investigate how the transcranial focused ultrasound influences the skull by using simulations and experiments to quantify the structural parameters and acoustic parameters. The manuscript is well written, and the structure is organized. The results between the experiments and simulations are consistent. There are no significant mistakes that need to be improved or revised. I would suggest accepting this manuscript in present form.   

Author Response

(The authors gave the same response as above.)

Reviewer 3 Report

This manucript presents some interesting results on skull model for transcranial ultrasound. The following comments may be addressed to strengthen the manuscript:

  1. Acrylic plate

The author selected an acrylic plate to mimic skull bone since the acoustic properties of the acrylic plate are close to those of skull based on the data from reference #18. However, the acoustic properties of bone and acryl are not very close to each other. Authors may compare the results with the published human skull data.

Moreover, the results show that the maximum acoustic pressure without an acrylic plate is about 550 kPa, while the maximum transcranial acoustic pressure is about 240 kPa. Given the utilized ultrasound frequency is high as 0.9 MHz, it is a quite low acoustic energy loss compared to published human skull data. Appropriate justifications should be provided.

  1. Page 3, Line 102.

It is not necessary to mention the information of hard disk.

  1. Page 4, Line 132.

The authors stated that “drive the transducer to generate continuous sinusoidal ultrasonic waves”. It is kind of confusing. Is that possible to drive an ultrasound transducer continuously? What is the material of this ultrasound transducer? Piezoelectric material?

  1. Page 4, Figure 2

Scale bar should be included in figure 2. (b).

5. Page 5, Line 168.

“Bone plate should be changed to “bone phantom” or “acrylic plate”.

6. Page 5, Figure 5.

Figure 5 (a) (b) (e) (f) should also include the unit of pressure in the colorbar.

7. Page 5, Figure 5 and Page 6, Table 2.

Should -6dB or -3dB be used? 

8. The fonts in Figures 7 -10 are too small to distinguish.

  1. Page 11, Line348.

The authors stated that “After comprehensive consideration, potential major reasons for the observed acoustic pressure field differences between self-programming and COMSOL include …”  The authors should explain what is the difference between Westervelt equation and Helmholtz equation, linearity and nonlinearity, FDTD and FEM, and how they influence the results?

10. The authors should double-check the grammar, typo, and vendor information. For example, there should be a space between the numerical value and the unit symbol.

Author Response

Dear reviewer:

On behalf of my co-authors, we thank you very much for giving us an opportunity to revise our manuscript, we appreciate editor and reviewers very much for their positive and constructive comments and suggestions on our manuscript. We have carefully analyzed your comments and have done our best to revise the manuscript based on them (important changes are highlighted in red), hoping to meet your expectations for our article.

We will be looking deeper into this direction and hope to communicate with you more in the future!

Good luck!

The following is an itemized response to your review comments:

[Comment 1]

Acrylic plate. The author selected an acrylic plate to mimic skull bone since the acoustic properties of the acrylic plate are close to those of skull based on the data from reference #18. However, the acoustic properties of bone and acryl are not very close to each other. Authors may compare the results with the published human skull data.

       Moreover, the results show that the maximum acoustic pressure without an acrylic plate is about 550 kPa, while the maximum transcranial acoustic pressure is about 240 kPa. Given the utilized ultrasound frequency is high as 0.9 MHz, it is a quite low acoustic energy loss compared to published human skull data. Appropriate justifications should be provided.

[Response]         Thanks to the reviewer for this suggestion. There are indeed some differences in the acoustic properties of acrylic plate and human skull. We compared the results of this paper with published data on human skulls, which are cited in the text (Page 5, Line 167-169), and found a high degree of agreement in the sound pressure attenuation results.

Mohammed noted a post-transcranial sound pressure attenuation of 56% compared to that in pure water. Ammi's study showed that the sound pressure attenuation after transcranial ultrasound at a frequency of 1.03 MHz ranged from 37.4% to 91.6%, with a mean attenuation of 64.68% in the five groups. These experimental data obtained using real human skulls are close to the results of this paper (56.22%, 57.65%, and 56.87%), which validates the reliability of the results of this paper to some extent.

The loss values in this paper are low relative to the acoustic energy loss of real human skull. We believe that this study ignores the inhomogeneous structure of the real skull such as voids and multilayers, and uses a homogeneous acrylic plate for simulation, so the obtained results will have relatively low acoustic energy loss. In the next study, we will try to conduct experiments using real human skulls to obtain more realistic results.

[Comment 2]

Page 3, Line 102. It is not necessary to mention the information of hard disk.

[Response]

Thanks to the reviewer for this suggestion. The unnecessary hard disk information has been removed from the article.

[Comment 3]

Page 4, Line 132. The authors stated that “drive the transducer to generate continuous sinusoidal ultrasonic waves”. It is kind of confusing. Is that possible to drive an ultrasound transducer continuously? What is the material of this ultrasound transducer? Piezoelectric material?

[Response]

Thanks to the reviewer for this suggestion. The original article was not written clearly here, so I'm sorry for the confusion it caused you. In this study, the continuous wave (CW) mode is used, and the power amplifier is used to continuously drive the transducer for 10s while measuring the acoustic field. Yes, the material of the ultrasonic transducer is the more commonly used piezoelectric ceramic. The expression of the transducer operation has been modified in the text (Page 3, Line 126), and the information on the transducer material has been added (Page 3-4, Line 129-132).

[Comment 4]

Page 4, Figure 2. Scale bar should be included in figure 2. (b).

[Response]

Thanks to the reviewer for this suggestion. The Scale bar has been increased in figure 2.(b).

[Comment 5]

Page 5, Line 168. “Bone plate should be changed to “bone phantom” or “acrylic plate”.

[Response]

Thanks to the reviewer for this suggestion. The updated manuscript has been corrected to “acrylic plate” (Page 5, Line 167).

[Comment 6]

Page 5, Figure 5. Figure 5 (a) (b) (e) (f) should also include the unit of pressure in the colorbar.

[Response]

Thanks to the reviewer for this suggestion. Figure 5 (a) (b) (e) (f) has been added to the unit information of sound pressure.

[Comment 7]

Page 5, Figure 5 and Page 6, Table 2. Should -6dB or -3dB be used? 

[Response]

Thanks to the reviewer for this suggestion. Strictly speaking, the half-attenuation point of the sound pressure should correspond to -6dB, and the Figure 5 and Table 2 have been corrected to -6dB.

[Comment 8]

The fonts in Figures 7 -10 are too small to distinguish.

[Response]

Thanks to the reviewer for this suggestion. Figures 7 -10 fonts have been reformatted and edited with enlarged font sizes to ensure readability.

[Comment 9]

Page 11, Line348. The authors stated that “After comprehensive consideration, potential major reasons for the observed acoustic pressure field differences between self-programming and COMSOL include …”  The authors should explain what is the difference between Westervelt equation and Helmholtz equation, linearity and nonlinearity, FDTD and FEM, and how they influence the results?

[Response]

Thanks to the reviewer for this suggestion. First, both Westervelt equation and Helmholtz equation are fluctuation equations. Westervelt equation is used to describe nonlinear propagation in many cases, while Helmholtz equation is mainly used to describe linear propagation (Page 11, Line 363-365). Then, acoustic phenomena are fundamentally nonlinear, and can be described by an approximate linearity when the ultrasound amplitude energy is low, and the nonlinear effect becomes progressively larger as the ultrasound amplitude increases (Page 11, Line 366-369). Finally, both FDTD and FEM are electromagnetic wave simulation algorithms. FDTD is faster in calculation and has average accuracy; FEM is average in calculation and has higher accuracy (Page 11-12, Line 371-373).

[Comment 10]

The authors should double-check the grammar, typo, and vendor information. For example, there should be a space between the numerical value and the unit symbol.

[Response] Thanks to the reviewer for this suggestion. We have thoroughly checked the article for grammar, typos and vendor information. Spaces have been added between all values and units.
